# Structural Homology-Based Drug Repurposing Approach for Targeting NSP12 SARS-CoV-2

**DOI:** 10.3390/molecules27227732

**Published:** 2022-11-10

**Authors:** Abdulelah Aljuaid, Abdus Salam, Mazen Almehmadi, Soukayna Baammi, Fahad M. Alshabrmi, Mamdouh Allahyani, Khadijah M. Al-Zaydi, Abdullah M. Izmirly, Sarah Almaghrabi, Bandar K. Baothman, Muhammad Shahab

**Affiliations:** 1Department of Clinical Laboratory Sciences, College of Applied Medical Sciences, Taif University, P.O. Box 11099, Taif 21944, Saudi Arabia; 2Precision Medicine Lab, Laboratory Building, Rehman Medical Institute, Phase-V, Hayatabad, Peshawar 25000, Khyber Pakhtunkhwa, Pakistan; 3African Genome Centre (AGC), Mohammed VI Polytechnic University, Benguerir 43150, Morocco; 4Department of Medical Laboratories, College of Applied Medical Sciences, Qassim University, Buraydah 51452, Saudi Arabia; 5Department of Chemistry, College of Science, University of Jeddah, Jeddah 23738, Saudi Arabia; 6Department of Medical Laboratory Sciences, Faculty of Applied Medical Sciences, King Abdulaziz University, P.O. Box 80216, Jeddah 21589, Saudi Arabia; 7Special Infectious Agents Unit-BSL3, King Fahd Medical Research Center and Medical Laboratory Technology Department, Faculty of Applied Medical Sciences, King Abdulaziz University, P.O. Box 80216, Jeddah 21589, Saudi Arabia; 8Center of Innovations in Personalized Medicine (CIPM), King Abdulaziz University, Jeddah 21589, Saudi Arabia; 9Department of Medical Laboratory Technology, Faculty of Applied Medical Sciences in Rabigh, King Abdulaziz University, Jeddah 21589, Saudi Arabia; 10State Key Laboratories of Chemical Resources Engineering, Beijing University of Chemical Technology, Beijing 100029, China

**Keywords:** drug repurposing, SARS-CoV-2, NSP-12, RdRp domain, FDA-approved antivirals

## Abstract

The severe acute respiratory syndrome coronavirus 2, also known as SARS-CoV-2, is the causative agent of the COVID-19 global pandemic. SARS-CoV-2 has a highly conserved non-structural protein 12 (NSP-12) involved in RNA-dependent RNA polymerase (RdRp) activity. For the identification of potential inhibitors for NSP-12, computational approaches such as the identification of homologous proteins that have been previously targeted by FDA-approved antivirals can be employed. Herein, homologous proteins of NSP-12 were retrieved from Protein DataBank (PDB) and the evolutionary conserved sequence and structure similarity of the active site of the RdRp domain of NSP-12 was characterized. The identified homologous structures of NSP-12 belonged to four viral families: *Coronaviridae*, *Flaviviridae*, *Picornaviridae*, and *Caliciviridae*, and shared evolutionary conserved relationships. The multiple sequences and structural alignment of homologous structures showed highly conserved amino acid residues that were located at the active site of the RdRp domain of NSP-12. The conserved active site of the RdRp domain of NSP-12 was evaluated for binding affinity with the FDA-approved antivirals, i.e., Sofosbuvir and Dasabuvir in a molecular docking study. The molecular docking of Sofosbuvir and Dasabuvir with the active site that contains conserved motifs (motif A-G) of the RdRp domain of NSP-12 revealed significant binding affinity. Furthermore, MD simulation also inferred the potency of Sofosbuvir and Dasabuvir. In conclusion, targeting the active site of the RdRp domain of NSP-12 with Dasabuvir and Sofosbuvir might reduce viral replication and pathogenicity and could be further studied for the treatment of SARS-CoV-2.

## 1. Introduction

Drug discovery is a time-consuming and costly process that involves a lengthy assessment of clinical trials and has a success rate of only 2%, with an average cost of USD 2–3 billion [1,2], and also fails to deliver at the time of pandemics [3]. On the other hand, drug repurposing has several advantages over drug discovery, and one is that it can be delivered in times of pandemics. At the beginning of February 2020, the World Health Organization (WHO) declared the novel coronavirus disease 2019 (COVID-19) a pandemic due to its rapid spread as well as its high mortality and morbidity rates [4]. COVID-19 first appeared in Wuhan city, China, in late 2019 and is still spreading all over the globe creating a disastrous effect on global health and the economy [5,6]. Due to its highly contagious nature, as of May 2022, the pandemic has spread to more than 200 countries and territories with over 5.5 million confirmed positive cases and mortality of over 350K across the globe [7].

The etiological agent of COVID-19 is severe acute respiratory syndrome coronavirus 2 (SARS-CoV-2), and it has been reported to originate from the animal coronavirus [8]. SARS-CoV-2 belongs to the genus Betacoronavirus of the family *Coronaviridae* which already has two other members, i.e., SARS coronavirus (SARS-CoV-1) and Middle East respiratory syndrome coronavirus (MERS-CoV) which caused pandemic onslaughts in 2002 and 2012, respectively, resulting in severe human losses both in terms of morbidity and mortality [9].

Coronaviruses are positive-sense single-stranded RNA with a genome size of ~30 kb containing a 5′ cap-structure and 3′-polyadenylated tail [10]. Their genome encodes for ORF1a and ORF1ab polyprotein which cleaves into multiple subunits and performs a diverse range of functions, i.e., binding and invading the host cell, viral replication, and evading the host immune system [11]. The proteins of coronaviruses perform their function by aggregating multiple subunits into an heteromeric complex, i.e., replication-transcription complex and several other structural proteins that assemble into new virions [12,13]. One of these subunits is a non-structural protein 12 (NSP-12) which involves RNA-dependent RNA polymerase (RdRp) activity.

NSP-12 of SARS-CoV-2 is highly conserved by sharing 96% similarity with SARS-CoV-1 and 70% with MERS-CoV. The recently resolved structure has also revealed conserved structural architecture by comparison with non-structural protein 5 (NS5) of flaviviruses and polymerases of picornaviruses [14]. However, NSP-12 has a length of ~930 amino acids in contrast to the polymerases of Human Poliovirus (PV) and NS5B of Hepatitis C Virus (HCV) which usually comprises ~500 amino acids. NSP-12 is responsible for viral replication and transcription once the virus invades into the host cellular environment by aggregating with cofactors such as NSP-7 and NSP-8 along with some additional subunits to initiate viral replication [15]. Its tertiary structure comprises three highly conserved domains: N-terminal nucleotidyltransferase (also known as nidovirus RdRp-associated nucleotidyltransferase (NiRAN)), interface, and C-terminal RdRp domain [14]. The RdRp domain of NSP-12 is highly conserved and responsible for viral replication and has been reported in several families of RNA genome viruses, while the NiRAN domain is only conserved among the members of nidoviruses by possessing kinase-like fold and involves in attachment with NSP-7 and NSP-8 [15]. RdRp domain contains seven conserved motifs (motif A-G) that involve attachment with nucleotide triphosphate (NTP) and RNA primer during the viral replication [16,17]. Due to these conserved evolutionary features of the RdRp domain, it has been suggested to be an important therapeutic target [18,19,20].

Currently, many studies including single-ligand docking [21,22] as well as screening of the compounds library [19,23] have been conducted to search for potential inhibitors against NSP-12 of SARS-CoV2. In our study, we employed the structural-based homology method to identify structurally-conserved proteins and then identified US Food and Drug Association (FDA)-approved inhibitors of the homologous structure. The selected FDA-approved inhibitors were Dasabuvir Sodium (Dasabuvir) [24] and Sofosbuvir [25] which were previously authorized for inhibiting the NS5B of HCV. Furthermore, the conserved active site of NSP-12 was characterized and the affinity of the inhibitors were evaluated using in silico assessments, i.e., molecular docking and simulations. 

## 2. Results

### 2.1. Structural Similarity-Based Searching and Screening for Homologous Proteins

The DALI server was employed for the identification of homologous protein structures to NSP-12 of SARS-CoV-2 that were already submitted in the PDB repository. The submitted query resulted in a total of 1105 homologous structures with a default cutoff of Z-score ≥2 as shown in Appendix A. For screening significant homologs, a cutoff Z-score value ≥19 was considered, resulting in 611 structures. These resulting homologous structures were involved in the RdRp activity and belonged to four viral families: *Coronaviridae*, *Flaviviridae*, *Picornaviridae*, and *Caliciviridae* and infected a diverse range of hosts. The representatives belonging to different viral genera were selected from the filtered homologous structures and their similarities were evaluated by using the similarity matrix and a cladogram by the neighbor joining method. The resulting matrix was visualized as a heatmap as shown in Figure 1A. The heatmap of Z-score-based structural similarity revealed the close relationship of coronaviruses with the members of *Flaviviridae* as well as *Caliciviridae* and *Picornaviridae*.

Based on structural similarity, the unrooted cladogram of homologous structures to NSP-12 was clustered into four viral families as shown in Figure 1B. The clade of the coronaviruses, i.e., SARS-CoV-1 and SARS-CoV-2, shares a close relationship with the NS5B of HCV and also with the other representatives of *Flaviviridae*, i.e., the Zika Virus (ZIKV), Dengue Virus (DENV), Bovine Viral Diarrhea Virus (BVDV), Japanese Encephalitis Virus (JEV), and Classical Swine Fever Virus (CSFV). The members of *Picornaviridae* included PolV, Human Coxsackievirus (CoxV), Human Rhinovirus (HRV), Human Enterovirus (HEV), Human Enterovirus 71 (HEV71), Foot and Mouth Disease Virus (FMDV), and Encephalomyocarditis virus. The representatives of *Caliciviridae* were the Sapporovirus (SapV), Rabbit Haemorrhagic Disease Virus (RHDV), Murine Norovirus (MNV), Norwalk Virus (NarV), and Human Norovirus (HNV). The clade of *Flaviviridae* and coronaviruses shared significant relationships as compared to the members of *Caliciviridae* and *Picornaviridae*.

### 2.2. Sequence and Structural Analysis of Conserved Motifs in NSP-12

The conserved motifs in NSP-12 were analyzed by multiple structures and sequence alignments implemented at the PROMALS3D server. The PROMLS3D server aligns multiple structures based on similarity at the amino acid sequence and secondary structure level. The identified conserved motifs in NSP-12 have been displayed in Figure 2A. NSP-12 has seven highly conserved motifs (motif A-G) which are responsible for NTP catalysis, binding to the 3′ of RNA template, and viral replication by agglomerating with NSP7 and NSP8 [14]. 

To display the conserved orientation of these motifs at the active site of NSP-12, multiple structures belonging to the family *Coronaviridae*, *Flaviviridae*, and *Picornaviridae* were superposed (structurally aligned) by using the MatchMaker algorithm implemented in UCSF Chimera. The conserved motifs identified from multiple sequence and structural alignment have been shown in Figure 2B. Motifs A–E are located in the palm domain while motifs F and G are present in the fingers domain of NSP-12. Motifs A and C have highly conserved aspartic acid residues which bind to the divalent ions (Mn^2+^ or Mg^2+^) and are responsible for executing the catalysis of nucleotide-binding [16,17]. Motifs B, D, E, and G are involved in nucleotide recognition and coordination. Motif F is enriched with positively charged basic residues which interact with the triphosphate moieties of the NTP [26,27]. Motif G has been also reported to involve in adjusting the orientation of the primer and template [28]. 

### 2.3. Pharmacophore Modeling and Druggable Site in NSP-12 

Pharmacophore modeling is a very useful technique for the detection of ligand-binding or a druggable site in protein that could be targeted to either trigger or to block the biological activity [29]. Pharmacophore modeling was computed by using the CAVITY search module of the CavityPlus server with a selection of “No Ligand” mode and default parameters. The CAVITY search module evaluates the pharmacophore or druggable cavity based on surface energy and site geometry. The analysis resulted in the prediction of a total of 27 cavities in NSP-12 and their information has been shown in Appendix A. The cavities were ranked based on factors, i.e., drug score, druggability, and predicted average PKD and the top-scored cavity as shown in Figure 2C was selected for further analysis.

The top-ranked cavity has a higher drug score revealing significant druggability of the active site with a predicted maximum pKd score of 6.99. The predicted maximum pKd score ≥6 indicates a suitable ligand ability of a cavity binding site. There was a total of 154 amino acid residues constituting the predicted cavity and they were compared with the evolutionarily conserved residues of the RdRp domain of NSP-12 from multiple sequence alignments. The comparison has been plotted as a Venn diagram as shown in Figure 3A. The highly conserved residues were K545 and R555 of motif F, D618, and D623 of motif A, S682, G683, T687, and N691 of motif B, D760, and D761 of motif C and W800 of motif D. These residues were further represented as a target in the RdRp domain of NSP-12 for molecular docking with the FDA-approved antivirals.

### 2.4. Molecular Docking of NSP-12 with FDA-Approved Antivirals

Before molecular docking, the ligands, i.e., Dasabuvir, Sofosbuvir, Ribavirin, GTP, and UTP, were prepared using AutoDock Tools. The grid box was centered at the active site or druggable cavity and the residues were considered rigid for both receptors, i.e., NSP-12 of SARS-CoV-2 and NS5B of HCV. The molecular docking was performed by AutoDock Vina with the parameters that are suggested for a single-docking experiment and the results for each abovementioned ligand were stored. AutoDock Vina predicts the docking score in kcal/mol and the lower energy means higher binding affinity between the ligand and the receptor. The results score of ligand bindings with the active site of NSP-12 and NS5B has been compared and visualized as a bar chart as shown in Figure 3B.

The docking score of NSP-12 with the ligands was in the range of −6.2 to −8.1 kcal/mol, while for NS5B, it was −6.5 to −9.1 kcal/mol, and both had an average difference of 0.84 kcal/mol. The average docking score of NSP-12 and NS5B for Sofosbuvir, Dasabuvir, Ribavirin, GTP, and GTP were −8.2, −8.6, −6.35, −8.25, and −7.6 kcal/mol, respectively. NSP-12 showed the lowest docking score (−8.1 kcal/mol) with Dasabuvir and higher (−6.2 kcal/mol) with Ribavirin. In general, the ligands showed a higher docking score for NS5B as compared to NSP-12. 

To examine the interactions between ligand and the conserved residues present at the active site of NSP-12, LigPlot + v2.1, standalone software was used. The predicted interactions of Sofosbuvir and Dasabuvir with NSP 12 have been shown in Figure 4A,B, respectively. The conserved residues involved in forming hydrogen bonds with Sofosbuvir were C622 of motif-A, W800 of motif-D, G811, and S814 of motif-E, while Dasabuvir interacted by the formation of hydrogen bonds with D761 of motif-C and W800 of motif-D. In addition, several conserved residues of motif-A, -C, and -E were also involved in hydrophobic interactions with Sofosbuvir and the residues of motif-A, -B, -C, -E, and -F with Dasabuvir. On the other hand, the native competitors such as GTP interacted with NSP-12 by forming hydrogen bonds with A554, R555, and T556 of motif-F, Y619, K621, and C622 of motif-A, while UTP interacted with S759, D760, and D761 of motif-C, W800 of motif-D, and G811 and S814 of motif-E. In the case of HCV as a receptor, Sofosbuvir binds with motif-B and -C, while Dasabuvir interacted with residues of motif-A, -B, -C, and motif-F, and their receptor–ligand interaction plots have been shown in Figure 4C,D, respectively. 

### 2.5. Molecular Dynamics (MD) Simulation

By binding to a protein’s functional binding cleft, each small molecule has the potential to cause significant conformational changes in the protein’s structure. To investigate the structural stability of 6M71, 100 ns of molecular dynamics (MD) simulations were run with the protein in both its apo and ligand-bound forms. The stability of the systems was evaluated using root mean square deviation (RMSD), root mean square fluctuation (RMSF), radius of gyration, and Hydrogen bond analysis.

#### 2.5.1. Root Mean Square Deviation (RMSD)

The average RMSD for 6M71, 6M71-Dasabuvir, Ribavirin, and Sofosbuvir complexes was calculated as 0.23, 0.23, 0.27, and 0.27 nm, respectively. All three systems’ RMSD values, which quantify conformational changes over time, indicated that the simulation was stable for up to 100 ns. The backbone RMSDs plot analysis revealed that the complex was stable throughout MD simulation despite a small deviation from its primary conformation. Figure 5 displays the RMSD plots of all complexes vs. simulated time. The resulting RMSD graph showed increasing trends with increasing RMSD values ranging from 0.10 to 0.33 nm between 0 and 60 ns in the case of 6M71/Ribavirin and 6M71/Sofosbuvir, suggesting the compounds were becoming accustomed to a new conformation inside the binding pocket; after that, the plateau continued and finally, settled at 0.29 nm, which does not exceed the 0.3 nm threshold. However, 6M71/Sofosbuvir has the same evolution to apo 6M71. Lower RMSD values for all complexes investigated shows that Dasabuvir inhibitor is stable inside 6M71 and gives a strong foundation for our research.

#### 2.5.2. Root Mean Square Fluctuation (RMSF)

The root mean square fluctuation (RMSF) method was used to investigate the effect of ligand binding on the flexible structure of protein as well as the behavior of essential amino acids. Throughout the simulation, a lower RMSF value indicates more rigidity, whereas a higher RMSF value indicates more flexibility. In Figure 6, RMSFs of the Apo NSP-12, NSP-12/Dasabuvir, NSP-12/Ribavirin, and NSP-12/Sofosbuvir complexes are displayed. When compared to the apo form of CAII, it ise shown that the fluctuations of the ligand-bound complexes’ residues are relatively stable, particularly at the region where the residues are engaged in ligand binding. Furthermore, the average RMSF values of Apo NSP-12, NSP-12/Dasabuvir, NSP-12/Ribavirin, and NSP-12/Sofosbuvir complexes were 0.10, 0.10, 0.11 and 0.11 nm, respectively. This finding implies that Dasabuvir, Ribavirin, and Sofosbuvir binding helped to maintain NSP-12 structurally stable.

#### 2.5.3. Radius of Gyration

We examined how binding of different ligands affected the protein’s overall compactness in its structure. For this goal, the radius of gyration (Rg) was calculated as a function of time. Any ligand with a high enough Rg value will be more likely to be flexible, making it unstable. On the other hand, lower Rg values point to confirmation of a dense and closely packed conformation. The average Rg values Apo NSP-12, NSP-12/Dasabuvir, NSP-12/Ribavirin, and NSP-12/Sofosbuvir complexes were found to be 2.99, 3.00, 2.97, 2.98 nm, respectively, suggesting that the binding of Dasabuvir, Ribavirin, and Sofosbuvir to NSP-12 packing did not cause significant change. Moreover, based on the graph (Figure 7), it appears that the Rg of NSP-12/Dasabuvir reached a stable equilibrium during the 100 ns simulation than NSP-12/Ribavirin and NSP-12/Sofosbuvir complexes. 

#### 2.5.4. Hydrogen Bonds Analysis

Further understanding of the examined chemicals’ molecular recognition, molecular interactions, and selectivity within the receptors can be attained using hydrogen bond analysis. For all MD trajectories of the complex, we calculated the number of hydrogen bonds created. Figure 8 displays the number of hydrogen bonds and pairs within 0.35 nm generated between NSP-12/Dasabuvir, NSP-12/Ribavirin, and NSP-12/Sofosbuvir complexes during the MD simulation. It was noticed that the Dasabuvir binds to the active pocket of NSP-12 with an average of 3.46 hydrogen bonds and 1.13 pairs within 0.35 nm. Similarly, Ribavirin acting as a ligand interacted with NSP-12 as a target within the binding site with an average of 3.97 hydrogen bonds, although the average of pairs within 0.35 nm is 1.66. However, the average of both hydrogen bonds and pairs within 0.35 nm for the NSP-12/Sofosbuvir complex was 1. On the other hand, the H-bonding plot showed that Dasabuvir was able to maintain a more robust interaction with the binding pockets of NSP-12 throughout the simulation time than Ribavirin and Sofosbuvir.

### 2.6. MMPBSA Binding Energy

Before MM/PBSA calculation, each complex between the receptors or inhibitors/ligands should reach the equilibrium state. From Figure 9, it is clear that the temperature of each complex quickly reaches the target value (300 K) and remains stable over 100 ns.

After that, we utilized a python script MmPbSaStat.py to calculate average free binding energy of the selected complexes (Table 1), as provided in the g_mmpbsa package. This script calculates the average free binding energy and its standard deviation/error from the output files, which were obtained from g_mmpbsa.

The energy liberated during the process of bond formation, or alternatively, the interaction between a ligand and protein, is shown in the form of binding energy. The lesser the binding energy, the better the binding of the ligand and protein. The final binding energy is a cumulative sum of van der Wall, electrostatic, polar solvation, and SASA energy. Except for the polar solvation energy, all other forms of energy contributed favorably to the interaction between different molecules (Dasabuvir, Ribavirin, and Sofosbuvir) and NSP-12. The bioactive molecule Dasabuvir showed the least binding free energy (−42.151 KJ/mol) among all the selected molecules. A comparison of the binding free energies of all the complexes was made by plotting the binding energy versus time graphs. Further, we examined the contribution of each residue of NSP-12 in terms of binding free energy to the interaction with the selected. The contribution of each residue was calculated by decomposing the total binding free energy of the system into per residue contribution energy molecules using python script MmPbSaDecomp.py provided by MM-PBSA package. From the overall results, it was observed that the amino acid residue Trp 800 showed the lowest contribution binding energy for all the tree complexes, suggesting their significant role in binding; indeed, contribution energy analysis of amino acid residues Tyr619, Cys622, Trp800, Glu811, Ser814, and Asp761 revealed that these amino acids played an important role during interaction, which confirms the docking result.

Finally, the Sofosbuvir and Dasabuvir could be established as a repurposed drug against the COVID-19 main protease.

## 3. Discussion

Drug repurposing or repositioning is a term coined for the study of therapeutic uses of existing drugs that are either previously approved or currently under the investigational phase. Drug repurposing offers potential advantages over the discovery of novel drugs, and one is that it can be delivered quickly without lengthy assessments and time-consuming clinical trials. Due to this significance, drug repurposing offers an effective way to save lives and time during pandemics. Many drugs have been already suggested for the current pandemic caused by SARS-CoV-2, of which the most prominent are FDA-approved Chloroquine and its derivatives—an anti-malarial drug and Remdesivir. Remdesivir, a nucleoside analog, has been shown to significantly reduce the SARS-CoV-2 pathogenicity in in vitro study [30] and is still under investigation [31]. However, there is still a need for the identification of more drugs that could be further investigated for efficiency in reducing viral pathogenicity by using in vitro studies. These drug searches can be improved by employing computational methods such as in silico screening and structure-based drug targeting methods. 

In this study, we searched for homologs of NSP-12 of SARS-CoV-2 and evaluated its binding affinity with the FDA-approved antivirals, i.e., Sofosbuvir and Dasabuvir, which specifically target the evolutionarily conserved site of NS5B of HCV [32]. The initial search for a homologous protein identified a total of 1105 structures by employing the DALI server. Further screening of these structures based on the quality of structural alignment (Z-score ≥ 19) resulted in 611 structures that belonged to four viral families, i.e., *Coronaviridae*, *Flaviviridae*, *Picornaviridae*, and *Caliciviridae*, responsible for RdRp activity during viral replication. The NSP-12 of SARS-CoV-2 displayed significant homology to the NS5 of *Flaviviridae* or NS5B of HCV as compared to the polymerases of *Picornaviridae* and *Caliciviridae*. Based on the pairwise alignment of amino acid sequences, NSP-12 shares 96% identity with SARS-CoV-1, 11% with HCV and HNV, and 10% with Poli. Despite sharing a lower identity based on sequence, NSP-12 has seven highly conserved motifs in the RdRp domain that play a key role in viral replication. These motifs have been previously reported to play essential roles in the viral replication of several viral families [33,34,35]. Motifs A–C have highly conserved residues while motifs D-G have conserved structural orientation at the active site of the RdRp domain.

The active site of RdRp was also identified as a suitable site for drug targeting based on the highest druggability score evaluated by using the CavityPlus webserver. The conserved residues and residues predicted from CavityPlus that were participating in ligand binding showed significant overlap when compared. The overlap included the conserved residues of motifs A, B, C, D, and F of the RdRp domain of NSP-12. These conserved residues have been also described previously to involve interactions with nucleoside, RNA primer attachment, and the RdRp activity [16]. The active site of the RdRp domain of NSP-12 was further evaluated for binding affinity with FDA-approved antivirals by the molecular docking method. Molecular docking was performed using AutoDock Vina which revealed a significant binding affinity of NSP-12 with Sofosbuvir and Dasabuvir as compared to Ribavirin. Despite that, NS5B showed higher binding affinity (lower binding energy) as compared to NSP-12; however, the differences were still insignificant. Further, MD simulations were carried out to determine the time dependent structural stability and intermolecular interactions. 

The profiling of NSP-12 and ligand interaction revealed firm binding due to the formation of hydrogen bonds and hydrophobic interactions. The participating conserved residues of motif-A, -B, -C, -D, -E, and -F of the RdRp domain were involved in these interactions with the ligands, i.e., Dasabuvir, Sofosbuvir, as well as with their native competitors, i.e., GTP and UTP. Similarly, in the case of HCV as a receptor, the residues of motif-A, -B, -C, and -F were involved in interactions with the above-mentioned ligands. In general, both Sofosbuvir and Dasabuvir showed higher binding to the active site of NS5B of HCV and NSP-12 of SARS-CoV-2. 

## 4. Material and Methods

### 4.1. Structural-Based Search for Homologous Proteins

We retrieved NSP-12 of SARS-CoV-2 with PDB ID: 6M71 at 2.9 Å resolution from the Protein Data Bank (PDB) repository that was recently submitted by Gao et al. [36]. For the identification of homologous structures in the PDB repository, 6M71 chain A was submitted to the DALI server (http://ekhidna2.biocenter.helsinki.fi/dali/ accessed on 16 March 2020). The DALI server identifies and ranks homologous protein structures based on the quality of structural alignment in the form of a Z-score [37]. The Z-score is a quantitative expression for comparing the intramolecular similarity between two homologous protein structures and its value ≥2 is suggested to be significant [38,39]. In this study, we used a Z-score cutoff of ≥19. To obtain a similarity matrix and a cladogram based on structural similarity, the same server was employed by using an “all against all” analysis for a total of 53 representative homologs. The heatmap was visualized by using the Pheatmap v1.0.12 package in R and the cladogram in FigTree v1.4.4 software.

### 4.2. Homologous Structures and Amino Acid Sequence Analysis

The selected homologous proteins responsible for RdRp activity were analyzed for conserved sequences (motifs) and secondary structure by employing the PROMALS3D server [40]. The resulting alignments were exported into ESPript 3.0 server for visualization (http://espript.ibcp.fr accessed on 5 April 2020). To further study the conserved residues at the active site of NSP-12, multiple structures were aligned by using the MatchMaker algorithm implemented in the University of California, San Francisco (UCSF) Chimera v.14 and visualized [41].

### 4.3. Pharmacophore Modeling and Prediction of Druggable Sites in NSP-12

For pharmacophore modeling or assessing druggable cavities in 6M71-A, the CavityPlus server (http://www.pkumdl.cn:8000/cavityplus/computation.php accessed on 12 April 2020) was employed by submitting the structure file. The CAVITY search module of the CavityPlus server was used to detect the ligand-binding cavity and druggability was predicted based on the energy of residues and site geometry with default parameters [42,43]. Those residues that were reported to be evolutionarily conserved in the RdRp domain as well as identified from druggable cavity search were visualized in the Venn diagram using Venny 2.1—an online server for plotting Venn diagram [44].

### 4.4. Preparation of Ligand and Receptor

AutoDock Tools v1.4.5 was employed for the ligand and receptor preparation [45]. In general, we followed the “Single-docking experiment with AutoDock Vina” method by following the guidelines as suggested by Forli et al. for ligand-receptor docking [46]. Initially, the Canonical SMILES IDs of the ligand compounds, i.e., Sofosbuvir (CID: 45375808), Ribavirin (CID: 37542), and Dasabuvir (CID: 56640146) were retrieved from the PubChem library [47] and then imported in Chimera v.14 for converting into PDB files. Subsequently, the structures were prepared by minimizing the free energy and addition of hydrogen atoms. On the other hand, the receptor structures were acquired from PDB and then processed in AutoDock Tools by removing water molecules, isolation of a single chain, addition of charges, merging non-polar hydrogens, and assigning appropriate atom types [46].

### 4.5. Molecular Docking

AutoDock Vina v1.1.2 was employed for molecular docking between ligand molecules and receptor [35]. A grid box of size 48 × 44 × 54 Å was selected and centered at the druggable cavity of 6M71-A oriented at 117.4, 115.3, 129.5 Å, while the NS5B of HCV (PDB ID: 1YUY-A) was considered as a control with the same size of the grid box used at the origin of 0.6, 80.0, and 64.2 Å. The parameters for molecular docking were used as described for running the single-docking experiment [46]. Furthermore, we chose the default scoring function implemented in AutoDock Vina [48]. The resulting score of molecular docking of NSP-12 of SARS-CoV-2 and NS5B of HCV were compared and plotted using a bar plot. For depicting protein–ligand interaction, LigPlot + v2.1 was used for characterization and visualization [49].

### 4.6. Molecular Dynamics (MD) Simulation

The MD simulation of the apo and ligand-bound complex of NSP-12 protein was done using the GROMACS v.2021 and GROMOS96 54a7 force field. The systems were solvated in a cubic box of TIP3P water model and neutralized with the addition of counter ions. The neutralized systems were minimized using the steepest descent algorithm with a maximum force of 1000 KJ mol^−1^ nm^−1^ to adjust the geometry. To obtain further convergence, the minimized systems were equilibrated for in an NVT ensemble followed by NPT ensemble. Finally, a 100ns production run was executed, with coordinates and energies being saved in the output trajectory file at 10ps intervals. The RMSD, RMSF, radius of gyration (Rg), and hydrogen bond contacts were used to determine the degree of convergence for all of the simulated systems.

### 4.7. MM-PBSA Calculation

The molecular mechanic/Poisson–Boltzmann surface area (MM-PBSA) methods were used to calculate of binding free energy NSP-12/Dasabuvir, NSP-12/Ribavirin, and NSP-12/Sofosbuvir complexes. This calculation validates the thorough ligand–protein interaction investigation. The binding energy constitutes potential energy, polar and non-polar solvation energies, and they were calculated by the ‘g_mmpbsa’ packages using GROMACS. The binding energy calculations in this method were calculated using the following equation:ΔG binding = G complex − (G receptor + G ligand)

The ΔG binding represents the total binding energy of the complex, while the binding energy of the free receptor is the G receptor, and that of unbounded ligand is represented by the G ligand.

## 5. Conclusions

Structural homology-based drug repurposing is a significant approach to exploring the homologous proteins which have been already targeted by either approved or investigational drugs. In this study, we searched for structural homologs of NSP-12 of SARS-CoV-2 and identified their relationship with the members of three viral families, i.e., *Flaviviridae*, *Picornaviridae*, and *Caliciviridae*. The RdRp domain of NSP-12 is highly conserved in terms of both sequence and architecture and assessed for binding affinity with FDA-approved antivirals, i.e., Dasabuvir and Sofosbuvir. These antivirals were initially approved for targeting the NS5B of HCV and serve as competitive inhibitors for binding to the active site that results in reducing viral pathogenesis. The results of this study revealed that the six conserved motifs (motif A-F) were involved in the formation of hydrogen bonds and hydrophobic interactions with the ligands. However, there is still a further need for in vitro analysis of these drugs to assess their efficiency before entering into clinical trials.

## Figures and Tables

**Figure 1 molecules-27-07732-f001:**
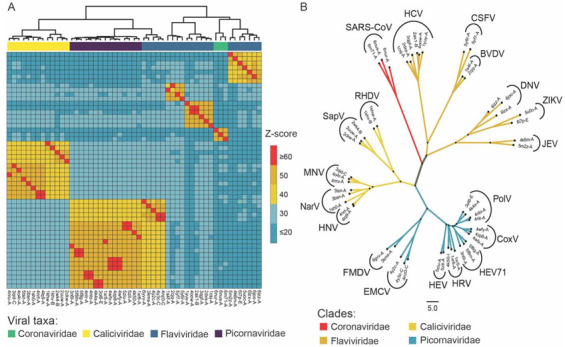
Comparative analysis of the homologous structures identified using the DALI server. (**A**) Heatmap of correlational similarity among homologous structures based on Z-score. (**B**) Unrooted tree of the homologous protein structures clustered into four viral families; *Coronaviridae*, *Caliciviridae*, *Flaviviridae*, and *Picornaviridae*.

**Figure 2 molecules-27-07732-f002:**
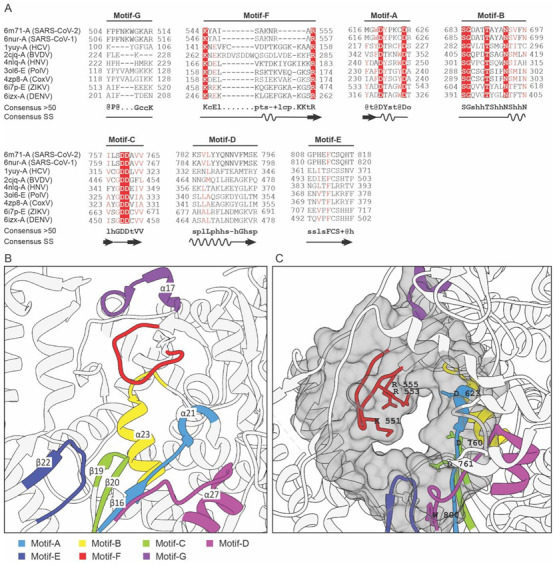
Analysis of the conserved motifs in NSP-12 and identification of druggable cavity. (**A**) Multiple sequences and secondary structure alignment of the homologous protein about NSP-12 of SARS-CoV-2 shows the conserved residues and secondary structure responsible for RdRp activity. Residues that have been highlighted in red are highly-conserved amino acids in the functional domain of RdRP protein. (**B**) Structurally and functionally conserved seven motifs A–G located at the active site of the RdRp domain of NSP-12. (**C**) The predicted druggable cavity at the active site of the RdRp domain of NSP-12. The predicted active residues participating in ligand interactions have been shown as a gray surface where the conserved residues are also located.

**Figure 3 molecules-27-07732-f003:**
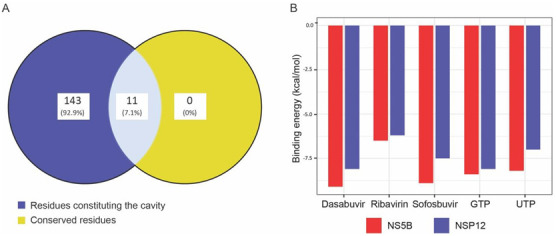
Comparison of conserved residues participating in ligand interactions and molecular docking score between NS5B and NSP-12. (**A**) A Venn diagram of active residues predicted from pharmacophore modeling and evolutionarily conserved residues of the RdRp domain of NSP-12. (**B**) Bar chart for comparison of binding energy score from AutoDock Vina between NSP-12 of SARS-CoV-2 and NS5B of HCV.

**Figure 4 molecules-27-07732-f004:**
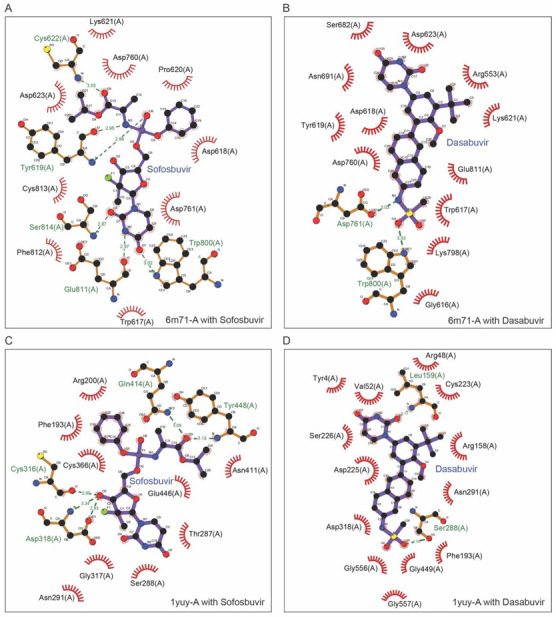
Receptor–ligand interaction profiling. The interactions between receptors, i.e., Nsp-12 and NS5B and ligands, i.e., Sofosbuvir and Dasabuvir. The dotted green line represents hydrogen bonding while the residues shown participate in hydrophobic interactions. (**A**) Interactions between NSP-12 and Sofosbuvir. (**B**) Interactions between NSP-12 and Dasabuvir. (**C**) Interactions between NS5B and Sofosbuvir. (**D**) Interactions between NS5B and Dasabuvir.

**Figure 5 molecules-27-07732-f005:**
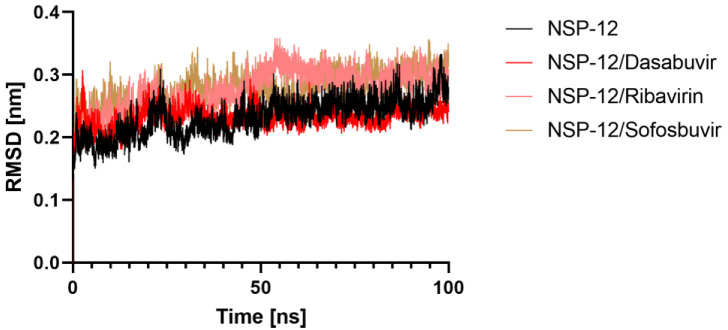
Structural stability analysis. RMSD of protein backbone vs. simulation time for solvated NSP-12 protein in complex with Dasabuvir, Ribavirin, and Sofosbuvir during 100 ns of molecular dynamics simulations.

**Figure 6 molecules-27-07732-f006:**
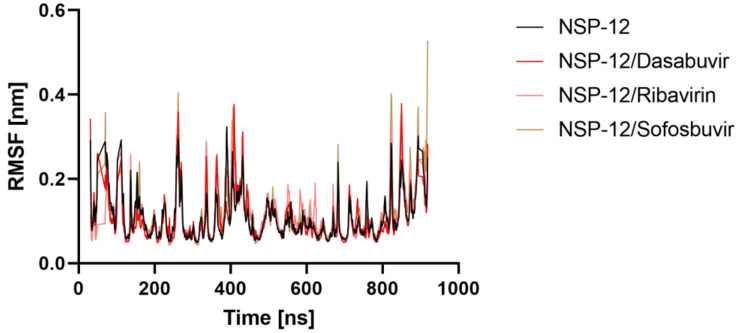
Ligand-induced flexibility. Root mean square deviation (RMSF) values of NSP-12 alone and in a complex with Dasabuvir, Ribavirin, and Sofosbuvir vs. the number of residues.

**Figure 7 molecules-27-07732-f007:**
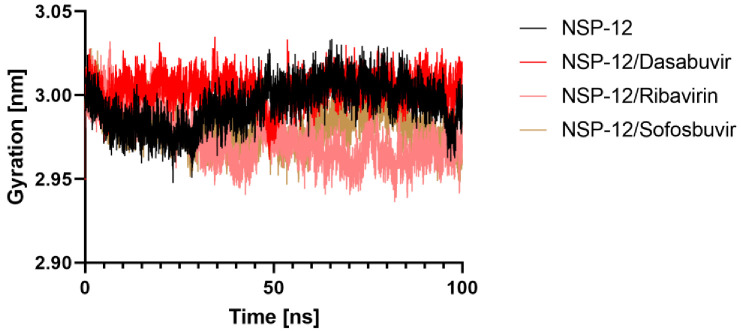
Structural compactness of NSP-12. The radius of gyration (Rg) for backbone atoms of NSP-12 alone and in complex with Dasabuvir, Ribavirin, and Sofosbuvir throughout the simulation.

**Figure 8 molecules-27-07732-f008:**
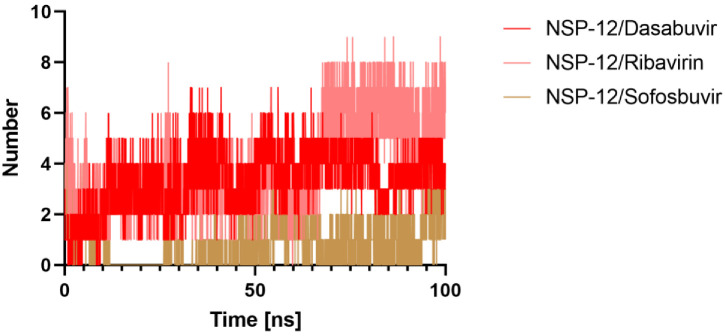
Intermolecular interactions. Hydrogen bonds numbers made between Dasabuvir, Ribavirin, and Sofosbuvir, with the NSP-12 protein active site residues during MD simulation.

**Figure 9 molecules-27-07732-f009:**
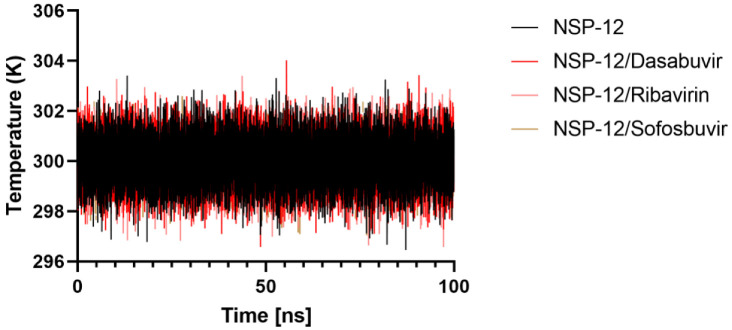
The system’s temperature during 100 ns of MD simulation.

**Table 1 molecules-27-07732-t001:** List of observed average and standard deviations of all energetic components including the binding energy taken from MM-PBSA analysis.

Complexes	Binding Energy (KJ/mol)	SASA Energy (KJ/mol)	Polar Solvation Energy (KJ/mol)	Electrostatic Energy (KJ/mol)	Van der Waal Energy (KJ/mol)
NSP-12/Dasabuvir	−42.151 ± 21.735	−20.615 ± 1.635	228.354 ± 25.854	−87.631 ± 20.339	−162.259 ± 16.736
NSP-12/Ribavirin	60.285 ± 28.431	−8.846 ± 1.638	420.099 ± 75.663	−327.013 ± 76.448	−23.956 ± 20.920
NSP-12/Sofosbuvir	−26.168 ± 54.225	−9.734 ± 5.282	94.581 ± 68.246	−26.753 ± 24.025	−84.262 ± 47.830

## Data Availability

Not applicable.

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
