# Peer review of "Structural Homology-Based Drug Repurposing Approach for Targeting NSP12 SARS-CoV-2"

_molecules, 2022, doi:10.3390/molecules27227732_

Round 1

Reviewer 1 Report

This manuscript is very well written, and the figures clearly presented. The authors provided proofs for structurally identified homologous could be used for drug repurposing based on the targets of the previously FDA-approved drugs. The authors provided an example of how such analysis could be done and this is an in silico approach on drug repurposing. The Materials and Methods are clearly written and the results are presented logically. The overall quality of the manuscript is excellent. The only drawback that I am thinking of is the chemicals that have been used for screening are  only limited to the three FDA approved chemicals plus GTP and UTP. I suggest expand the chemical pool for additional screening.

Reviewer 2 Report

Comments to the authors:

1. Abstract: Authors need to re-write the abstract and include results in numeric values from their analysis.

2. There are thousands of studies related to drug repurposing against SARS-CoV-2. So, how authors' work is different from other published works?

3. Introduction Page No 2 Line 91….95 Authors stated, "Recently, several drugs have been suggested for ‘reducing the SARS-CoV-2 pathogenicity by targeting the active site of the NSP-12 [2]. One of these drugs is the currently investigational Remdesivir – a nucleoside analog recently shown to target the NSP-12 and reduces viral replication in both in vitro and clinical studies [21, 22]. However, no single study is available that could propose authorized drugs by utilizing structure-based drug repurposing approaches”.

Authors need to avoid such claims that there are no other studies. Drug repurposing is a widespread technique for researchers targeting different viral molecules to repurpose FDA-approved drugs.

Example: ncbi.nlm.nih.gov/pmc/articles/PMC8282484/

https://www.mdpi.com/1999-4915/14/6/1345/pdf

https://www.intechopen.com/online-first/82800

4. Section 2.1 Line no 104 Authors selected PDB ID: 6M71. In Protein Data Bank, there are several structures of receptor molecules. What were your selection criteria for choosing PDB ID: 6M71? I need to clarify and mention this in the manuscript.

5. Section 2.2 How have the authors identified the active site of NSP-12? I must clarify and mention it in the main text with suitable references.

6. In section 2.1 authors used the DALI server, and in section 2.2 MatchMaker algorithm inbuild in the Chimera tool. What was the specific reason for using two different software for the same purpose? Need to be clarified.

7.ESPript 3.0 server and Chimera could be utilized for alignment visualization. Why did the authors use two different software? What specific reason needs to be clarified?

8. Section 2.4

Authors need to predict active sites of molecules  6M71 and 1YUY, then further docking and MDS should perform.

9. Which scoring function was used for docking in the AutoDock tool? Need to be clarified and mentioned in the manuscript.

10. Add citations and references for the AutoDock tool.

11. How authors have prepared Receptor and drug molecules for docking is not clear.

Need to clarify how authors minimized the receptor molecules?

12. Why did the authors not mention the preparation of drug molecules?

13. Add citation/reference for LigPlus v2.1.

14. Correct citation of each software, tool, and database needs to incorporate into the manuscript.

15. Authors should plot all MDS plots for all molecules on a single plot for better comparison visualization for RMSD, RMSF, Radius of Gyration, Hydrogen bond, MMPBSA energy data, etc. For Example, there should be only 1 RMSD plot for all complexes.

16. “Figure 8. Intermolecular interactions. Hydrogen bonds numbers made between (A) Dasaburiv, (B) 353 Ribavirin, and (C) Sofosbuvir, in the NSP-12 protein active site residues during MD simulation” What is pairs within 0.35 nm” with red color bars. Need to clarify and correct it accordingly.

17. Authors performed 100ns MDS for best complexes, including control (Receptor and Ligand in the presence of control data). It is recommended to analyze MM/GBSA or MM/PBSA calculations and represent that data with suitable plots like RMSD, Radius of gyration, RMSF, and Hydrogen bond analysis. Energy contribution plots should be generated and incorporated into the manuscript for MM/GBSA or MM/PBSA per amino acid residue. Authors may refer to the following manuscript for more details about MDS data representation (https://www.ncbi.nlm.nih.gov/pmc/articles/PMC8043163/)

18. Before ΔG of configurations was further calculated by the MM/GBSA or MM/PBSA method, the system about complex between the receptors or inhibitors/ligands should reach the equilibrium state. Complete details (such as changes in total temperature or energy over time) should be included and presented to verify that the system has well equilibrated.

19. No need to provide such a large supplementary file. If authors want to add additional files, they can provide MDS data as a supplementary file.

20. Grammar/English/Typo errors check required.

Round 2

Reviewer 2 Report

The authors have completed the suggested modifications and answered raised queries.